# Robust Loss Functions for Complementary Labels Learning

## Abstract

In ordinary-label learning, the correct label is given to each training sample. Similarly, a complementary label is also provided for each training sample in complementary-label learning. A complementary label indicates a class that the example does not belong to. Robust learning of classifiers has been investigated from many viewpoints under label noise, but little attention has been paid to complementary-label learning. In this paper, we present a new algorithm of complementary-label learning with the robustness of loss function. We also provide two sufficient conditions on a loss function so that the minimizer of the risk for complementary labels is theoretically guaranteed to be consistent with the minimizer of the risk for ordinary labels. Finally, the empirical results validate our method's superiority to current state-of-the-art techniques. Especially in cifar10, our algorithm achieves a much higher test accuracy than the gradient ascent algorithm, and the parameters of our model are less than half of the ResNet-34 they used.

## 1 Instruction

Deep neural networks have exhibited excellent performance in many real-applications. Yet, their supper performance is based on the correctly labeled large-scale training set. However, labeling such a large-scale dataset is time-consuming and expensive. For example, the crowd-workers need to select the correct label for a sample from 100 labels for CIFAR100. To migrate this problem, reachers have proposed many solutions to learn from weak-supervision: Noise-label learning Li et al. (2017); Hu et al. (2019); Lee et al. (2018); Xia et al. (2019), semi-supervised learning Zhai et al. (2019); Berthelot et al. (2019); Rasmus et al. (2015); Miyato et al. (2019); Sakai et al. (2017), similar-unlabeled learning Tanha (2019); Bao et al. (2018); Zelikovitz & Hirsh (2000), unlabeled-unlabeled learning Lu et al. (2018); Chen et al. (2020a;b), positive-unlabeled learning Elkan & Noto (2008); du Plessis et al. (2014); Kiryo et al. (2017), contrast learning Chen et al. (2020a;b), partial label learning Cour et al. (2011); Feng & An (2018); Wu & Zhang (2018) and others.

We investigate complementary-label learning Ishida et al. (2017) in this paper. A complementary Label is only indicating that the class label of a sample is incorrect. In the view of label noise, complementary labels can also be viewed as noise labels but without any true labels in the training set. Our task is to learn a classifier from the given complementary labels, predicting a correct label for a given sample. Collecting complementary labels is much easier and efficient than choosing a true class from many candidate classes precisely. For example, the label-system uniformly chooses a label for a sample. It has a probability of $\frac{1}{k}$ to be ordinary-label but $\frac{k-1}{k}$ to be complementary-label. Moreover, another potential application of complementary-label is data privacy. For example, on some privacy issues, it is much easier to collect complementary-label than ordinary-label.

Robust learning of classifiers has been investigated from many viewpoints in the presence of label noise Ghosh et al. (2017), but little attention paid to complementary-label learning. We call a loss function robust if the minimizer of risk under that loss function with complementary labels would be the same as that with ordinary labels. The robustness of risk minimization relies on the loss function used in the training set.

This paper presents a general risk formulation that category cross-entropy loss (CCE) can be used to learn with complementary labels and achieve robustness. We then offer some innovative analytical results on robust loss functions under complementary labels. Having robustness of risk minimization

helps select the best hyper-parameter by empirical risk since there are no ordinary labels in the validation set. We conclude two sufficient conditions on a loss function to be robust for learning with complementary labels. We then explore some popular loss functions used for ordinary-label learning, such as CCE, Mean square error (MSE) and Mean absolute error (MAE), and show that CCE and MAE satisfy our sufficient conditions. Finally, we present a learning algorithm for learning with complementary labels, named exclusion algorithm. The empirical results well demonstrate the advantage of the theoretical results we addressed and verify our algorithm's superiority to the current state-of-the-art methods. The contribution of this paper can be summarized as:

- We present a general risk formulation that can be view as a framework to employing a loss function that satisfies our robustness sufficient condition to learn from complementary labels.

- We conclude two sufficient conditions on a loss function to be robust for learning with complementary labels.

- We prove that the minimizer of the risk for complementary labels is theoretically guaranteed to be consistent with the minimizer of the risk for ordinary labels.

- The empirical results validate the superiority of our method to current state-of-the-art methods.

## 2 RELATED WORKS

Complementary-label refers to that the pattern does not belong to the given label. Learning from complementary labels is a new topic in supervised-learning. It was first proposed by Ishida et al. (2017). They conduct such an idea to try to deal with time-consuming and expensive to tag a large-scale dataset.

In their early work Ishida et al. (2017), they assume the complementary labels are the same probability to be selected for a sample. And then, based on the ordinary one-versus-all (OVA) and pairwise-comparison (PC) multi-class loss functions Zhang (2004) proposed a modifying loss for learning with complementary labels.

Even though they provided theoretical analysis with a statistical consistency guarantee, the loss function met a sturdy restriction that needs to be symmetric ($\ell(z) + \ell(-z) = 1$). Such a severe limitation allows only the OVA and PC loss functions with symmetric non-convex binary losses. However, the categorical cross-entropy loss widely used in the deep learning domain, can not be employed by the two losses they defined.

Later, Yu et al. (2018a) assume there are some biased amongst the complementary labels and presents a different formulation for biased complementary labels by using the forward loss correction technique Patrini et al. (2017) to modify traditional loss functions. Their suggested risk estimator is not necessarily unbiased and proved that learning with complementary labels can theoretically converge to the optimal classifier learned from ordinary labels based on the estimated transition matrix. However, the key to the forward loss correction technique is to evaluate the transition matrix correctly. Hence, one will need to assess the transition matrix beforehand, which is relatively tricky without strong assumptions. Moreover, in such a setup, it restricts a small complementary label space to provide more information. Thus, it is necessary to encourage the worker to provide more challenging complementary labels, for example, by giving higher rewards to the specific classes. Otherwise, the complementary label given by the worker may be too evident and uninformative. For example, class three and class five are not class one evidently but is uninformative. This paper focuses on the uniform (symmetric) assumption and study random distribution as a biased assumption (asymmetric or non-uniform).

Based on the uniform assumption, Ishida et al. (2019) proposed an unbiased estimator with a general loss function for complementary labels. It can make any loss functions available for use, not only soft-max cross-entropy loss function, but other loss functions can also be utilized. Their new framework is a generalization of previous complementary-label learning Ishida et al. (2017). However, their proposed unbiased risk estimator has an issue that the classification risk can attain negative values after learning, leading to overfitting Ishida et al. (2019). They then offered a non-negative correction to the original unbiased risk estimator to improve their estimator, which is no longer

guaranteed to be an unbiased risk estimator. In this paper, our proposed risk estimator is also not unbiased, but the minimizer of the risk for complementary labels is theoretically guaranteed to be consistent with the minimizer of the risk for ordinary labels, both uniform and non-uniform.

## 3 PRELIMINARIES

### 3.1 LEARNING WITH ORDINARY LABELS

In the context of learning with ordinary labels, let $\mathcal{X} \subset \mathbb{R}^d$ be the feature space and $\mathcal{Y} = \{1, \cdots, k\}$ be the class labels. A multi-class loss function is a map: $\mathcal{L}(f_\theta(\boldsymbol{x}), y) : \mathcal{X} \times \mathcal{Y} \to \mathcal{R}^+$. A classifier can be presented as:

$$h(\boldsymbol{x}) = \arg\max_{i \in [k]} f_\theta^{(i)}(\boldsymbol{x}) , \tag{1}$$

where $f_\theta(\boldsymbol{x}) = (f_\theta^{(1)}(\boldsymbol{x}), \cdots, f_\theta^{(k)}(\boldsymbol{x}))$, $\theta$ is the set of parameters in the CNN network, $f_\theta^{(i)}(\boldsymbol{x})$ is the probability prediction for the corresponding class $i$. Even though $h(x)$ is the final classifier, we use notation of calling $f_\theta(\boldsymbol{x})$ itself as the classifier. Given dataset $\mathcal{S} = \{(\boldsymbol{x}_i, y_i)\}_i^N$, together with a loss function $\mathcal{L}$, $\forall f_\theta \in \mathcal{F}$ ($\mathcal{F}$ is the function space for searching), $\mathcal{L}$-risk is defined as:

$$\mathcal{R}_\mathcal{L}^\mathcal{S}(f_\theta) = \mathbb{E}_\mathcal{D}\left[\mathcal{L}(f_\theta(\boldsymbol{x}), y)\right] = \mathbb{E}_\mathcal{S}\left[\mathcal{L}(f_\theta(\boldsymbol{x}), y)\right], \tag{2}$$

Some popular multi-class loss functions are CCE, MAE, MSE. Specifically,

$$\ell(f_\theta(\boldsymbol{x}), y) = \ell(\mathbf{u}, y) = \begin{cases} \sum_{i=1}^k \mathbf{e}_y^{(i)} \log \frac{1}{\mu_y} = \log \frac{1}{\mu_y} & \text{CCE,} \\[2mm] \|\mathbf{e}_y - \mathbf{u}\|_1 = 2 - 2\mu_y & \text{MAE,} \\[2mm] \|\mathbf{e}_y - \mathbf{u}\|_2^2 = \|\mathbf{u}\|_2^2 + 1 - 2\mu_y & \text{MSE,} \end{cases} \tag{3}$$

where $\mathbf{u} = f_\theta(\boldsymbol{x}) = (\mu_1, \cdots, \mu_k)$, and $\mathbf{e}_y$ is a one-hot vector that the $y$-th component equals to 1, others are 0. The goal of multi-class classification is to learn a classifier $f_\theta(\boldsymbol{x})$ that minimize the classification risk $\mathcal{R}_\mathcal{L}^\mathcal{S}$ with multi-class loss $\mathcal{L}$.

### 3.2 LEARNING WITH COMPLEMENTARY LABELS

In contrast to the ordinary-label learning, the complementary-label (CL) dataset contains only labels indicating that the class label of a sample is incorrect. Corresponding to the ordinary labels dataset $\mathcal{S}$, the independent and identically distributed (i.i.d.) complementary labels dataset denoted as:

$$\bar{\mathcal{S}} = \{(\boldsymbol{x}, \bar{y})\}_i^N, \tag{4}$$

where $N$ is the size of the dataset $\bar{\mathcal{S}}$, and $\bar{y}$ represents that pattern $\boldsymbol{x}$ does not belong to class-$\bar{y}$.

The general labels' distribution of dataset $\bar{\mathcal{S}}$ is as:

$$P(\bar{y}|y) = \begin{bmatrix} 0 & p_{12} & \dots & p_{1k} \\ p_{21} & 0 & \dots & p_{2k} \\ \vdots & \vdots & \ddots & \vdots \\ p_{k1} & \dots & p_{k(k-1)} & 0 \end{bmatrix}_{k \times k,} \tag{5}$$

where $p_{ij}$ denotes that the probability of the $i$-th class's pattern $\boldsymbol{x}$ labeled as $j$, $\sum_{j=1}^k p_{ij} = 1, p_{ij \neq 0}, j \neq i$. Supposing that the label system uniformly select a label from $\{1, \cdots, k\} \setminus \{y\}$ for each sample $\boldsymbol{x}$, then the Eq. (5) becomes

$$P(\bar{y}|y) = \begin{bmatrix} 0 & \frac{1}{k-1} & \dots & \frac{1}{k-1} \\ \frac{1}{k-1} & 0 & \dots & \frac{1}{k-1} \\ \vdots & \vdots & \ddots & \vdots \\ \frac{1}{k-1} & \dots & \frac{1}{k-1} & 0 \end{bmatrix}_{k \times k .} \tag{6}$$

Yu et al. (2018b) make a strong assumption that there are some bias in Eq. (5), while Ishida et al. (2017; 2019) focus on the assumption of Eq. (6). In this paper, we study both kinds of distribution.

## 4 METHODOLOGY

In this section, we firstly propose a general risk formulation for leaning with complementary labels. And then prove that some loss functions designed for the ordinary labels learning are robust to complementary labels with our risk formulation, such as categorical cross-entropy loss and mean absolute error.

### 4.1 GENERAL RISK FORMULATION

The goal of learning with complementary labels is to learn a classifier that predicts a correct label for any sample drawn from the same distribution. Because there are not ordinary labels for the model, we need to design a loss function or model for learning with complementary labels. The key to learning a classifier for ordinary label learning is to maximize the true label's predict-probability. One intuitive way to maximize the true label's predict-probability is to minimize the predict-probability of complementary labels. In this paper, with little abuse of notation, we let

$$
\begin{aligned}
\mathbf{u} &= f_\theta(\boldsymbol{x}) = (\mu_1, \cdots, \mu_k) \\
\mathbf{v} &= \mathbf{1} - f_\theta(\boldsymbol{x}) = (1 - \mu_1, \cdots, 1 - \mu_k) \ .
\end{aligned}
\tag{7}
$$

**Definition 1.** *(CL-loss) Together with loss function $\ell$ designed for the ordinary-label learning, the loss for learning with complementary-label is as:*

$$
\bar{\ell}(f_\theta(\boldsymbol{x}), \bar{y}) = \bar{\ell}(\mathbf{u}, \bar{y}) = \ell(\mathbf{v}, \bar{y}) \ .
\tag{8}
$$

### 4.2 THEORETICAL RESULTS

**Definition 2.** *(Robust loss) In the framework of risk minimization, a loss function is called robust loss function if minimizer of risk with complementary labels would be the same as with ordinary labels, i.e.,*

$$
\mathcal{R}_{\bar{\ell}}^{\bar{\mathcal{S}}}(f_{\theta^*}) - \mathcal{R}_{\bar{\ell}}^{\bar{\mathcal{S}}}(f_\theta) \le 0 \Rightarrow \mathcal{R}_\ell^{\mathcal{S}}(f_{\theta^*}) - \mathcal{R}_\ell^{\mathcal{S}}(f_\theta) \le 0 \ .
\tag{9}
$$

**Theorem 1.** *Together with $\ell$, $\bar{\ell}$ is a robust loss function for learning with complementary labels, if $\bar{\ell}$ satisfies:*

$$
\frac{\partial \bar{\ell}(\mathbf{u}, \bar{y})}{\partial \mu_{\bar{y}}} > 0, \ \frac{\partial \bar{\ell}(\mathbf{u}, \bar{y})}{\partial \mu_i} = 0, \ \forall i \in \{1, \cdots, k\} \setminus \{\bar{y}\} \ .
\tag{10}
$$

Note that, in Eq. 10, it means that $\bar{\ell}$ is a monotone increasing loss function **only on** $\mathbf{u}^{(\bar{y})}$.

*Proof.* Recall that for any $f_\theta$, and any $\ell$,

$$
\mathcal{R}_\ell^{\mathcal{S}}(f_\theta) = \mathbb{E}_{(\boldsymbol{x}, y)} \left[ \ell \left( f_\theta(\boldsymbol{x}), y \right) \right] = \frac{1}{|\mathcal{S}|} \sum_{(\boldsymbol{x}, y) \in \mathcal{S}} \ell \left( f_\theta(\boldsymbol{x}), y \right) \ .
\tag{11}
$$

For any complementary-label distribution in Eq. (5), and any loss function $\ell$, we have

$$
\begin{aligned}
\mathcal{R}_{\bar{\ell}}^{\bar{\mathcal{S}}}(f_\theta) &= \mathbb{E}_{(\boldsymbol{x}, \bar{y})} \left[ \bar{\ell} \left( f_\theta(\boldsymbol{x}), \bar{y} \right) \right] \\
&= \frac{1}{|\bar{\mathcal{S}}|} \sum_{i=1}^k \sum_{\boldsymbol{x} \in \mathcal{S}_i} \sum_{j \ne i}^k p_{ij} \bar{\ell} \left( f_\theta(\boldsymbol{x}), j \right),
\end{aligned}
\tag{12}
$$

where $p_{ij}$ is the component of complementary labels distribution matrix $P$, $\mathcal{S}_1 \cup \cdots \cup \mathcal{S}_k = \mathcal{S}$.

Supposing that $f_{\theta^*}$ is the optimal classifier learns from the complementary labels, and $\forall f \in \mathcal{F}$, where $\mathcal{F}$ is the function space for searching, we have

$$
\mathcal{R}_{\bar{\ell}}^{\bar{\mathcal{S}}}(f_{\theta^*}) - \mathcal{R}_{\bar{\ell}}^{\bar{\mathcal{S}}}(f_\theta) = \frac{1}{|\bar{\mathcal{S}}|} \sum_{i=1}^k \sum_{\boldsymbol{x} \in \mathcal{S}_i} \sum_{j \ne i}^k p_{ij} \left( \bar{\ell} \left( f_{\theta^*}(\boldsymbol{x}), j \right) - \bar{\ell} \left( f_\theta(\boldsymbol{x}), j \right) \right) \le 0,
\tag{13}
$$

where $p_{ij} \ne 0$. If $\exists \boldsymbol{x}' \in \bar{\mathcal{S}}$, s.t., $\bar{\ell}(f_{\theta^*}(\boldsymbol{x}'), \bar{y}) > \bar{\ell}(f_\theta(\boldsymbol{x}'), \bar{y})$, let $f_{\theta'}$ satisfying

$$
f_{\theta'}(\boldsymbol{x}) = \begin{cases} f_{\theta^*}(\boldsymbol{x}) & \boldsymbol{x} \in \bar{\mathcal{S}} \setminus \{\boldsymbol{x}'\}, \\ f_\theta(\boldsymbol{x}) & \boldsymbol{x} = \boldsymbol{x}', \end{cases}
\tag{14}
$$

then according to Eq. 12 and 13, $\mathcal{R}_{\bar{\ell}}^{\bar{\mathcal{S}}}(f_{\theta'}) < \mathcal{R}_{\bar{\ell}}^{\bar{\mathcal{S}}}(f_{\theta^*})$, $f_{\theta^*}$ is not the optimal classifier. This contradicts the hypothesize that $f_{\theta^*}$ is the optimal classifier.

Thus, $\forall \bar{y} \in \{1, \cdots, k\} \setminus \{y\}$, we have

$$\bar{\ell}(f_{\theta^*}(\boldsymbol{x}), \bar{y}) \leq \bar{\ell}(f_{\theta}(\boldsymbol{x}), \bar{y}) . \tag{15}$$

According to Eq. (10), $\bar{\ell}$ is a monotone increasing loss function **only on $\mathbf{u}^{(\bar{y})}$**, then we have

$$\forall \bar{y} \in \{1, \cdots, k\} \setminus \{y\}, \ f_{\theta^*}^{(\bar{y})}(\boldsymbol{x}) \leq f_{\theta}^{(\bar{y})}(\boldsymbol{x}) . \tag{16}$$

Thus,

$$f_{\theta^*}^{(y)}(\boldsymbol{x}) \geq f_{\theta}^{(y)}(\boldsymbol{x}), \ \left( f_{\theta}^{(y)}(\boldsymbol{x}) = 1 - \sum_{\bar{y} \neq y} f_{\theta}^{(\bar{y})}(\boldsymbol{x}) \right) \tag{17}$$

and then,

$$\ell(f_{\theta^*}(\boldsymbol{x}), y) \leq \ell(f_{\theta}(\boldsymbol{x}), y), \tag{18}$$

thus,

$$\mathcal{R}_{\ell}^{\mathcal{S}}(f_{\theta^*}) - \mathcal{R}_{\ell}^{\mathcal{S}}(f_{\theta}) \leq 0 . \tag{19}$$

$\square$

**Theorem 2.** *Together with $\ell$, $\bar{\ell}$ is a robust loss function for learning with complementary labels under symmetric distribution or uniform distribution, if $\bar{\ell}$ satisfies:*

$$\frac{\partial \bar{\ell}(\mathbf{u}, \bar{y})}{\partial \mu_{\bar{y}}} > 0, \ \sum_{i=1}^{k} \bar{\ell}(\mathbf{u}, i) = C, \ \text{(C is a constant)} . \tag{20}$$

It should be noted that, in Eq. 20, it means that $\bar{\ell}$ is a symmetric loss ($\sum \ell(\mathbf{u}, i) = C$), and $\bar{\ell}$ is a monotone increasing loss function on any $\bar{y}$.

*Proof.* For any complementary-label distribution in Eq. (6), and any loss function $\ell$, we have

$$\begin{aligned}
\mathcal{R}_{\bar{\ell}}^{\bar{\mathcal{S}}}(f_{\theta}) &= \mathbb{E}_{(\boldsymbol{x}, \bar{y})} \left[ \bar{\ell}(f_{\theta}(\boldsymbol{x}), \bar{y}) \right] \\
&= \frac{1}{|\bar{\mathcal{S}}|} \sum_{i=1}^{k} \sum_{\boldsymbol{x} \in \mathcal{S}_i} \sum_{j \neq i}^{k} \frac{1}{k-1} \bar{\ell}(f_{\theta}(\boldsymbol{x}), j) \\
&= \frac{1}{|\bar{\mathcal{S}}|} \sum_{i=1}^{k} \sum_{\boldsymbol{x} \in \mathcal{S}_i} \frac{1}{k-1} \left( C - \bar{\ell}(f_{\theta}(\boldsymbol{x}), i) \right) \\
&= \frac{C}{k-1} - \mathcal{R}_{\bar{\ell}}^{\mathcal{S}}(f_{\theta}),
\end{aligned} \tag{21}$$

where $\mathcal{S}_1 \cup \cdots \cup \mathcal{S}_k = \mathcal{S}$ .

Supposing that $f_{\theta^*}$ is the optimal classifier learns from the complementary labels, and $\forall f \in \mathcal{F}$, where $\mathcal{F}$ is the function space for searching, we have

$$\mathcal{R}_{\bar{\ell}}^{\bar{\mathcal{S}}}(f_{\theta^*}) - \mathcal{R}_{\bar{\ell}}^{\bar{\mathcal{S}}}(f_{\theta}) = \mathcal{R}_{\bar{\ell}}^{\mathcal{S}}(f_{\theta}) - \mathcal{R}_{\bar{\ell}}^{\mathcal{S}}(f_{\theta^*}) \leq 0, \tag{22}$$

According to the first constraint in Eq. (20), we then have

$$\bar{\ell}(f_{\theta}(\boldsymbol{x}), y) \leq \bar{\ell}(f_{\theta^*}(\boldsymbol{x}), y), \ \left( f_{\theta}^{(y)}(\boldsymbol{x}) \leq f_{\theta^*}^{(y)}(\boldsymbol{x}) \right) \tag{23}$$

and then,

$$\ell(f_{\theta^*}(\boldsymbol{x}), y) \leq \ell(f_{\theta}(\boldsymbol{x}), y), \tag{24}$$

thus,

$$\mathcal{R}_{\ell}^{\mathcal{S}}(f_{\theta^*}) - \mathcal{R}_{\ell}^{\mathcal{S}}(f_{\theta}) \leq 0 . \tag{25}$$

$\square$

---

**Algorithm 1** Learning from complementary labels by exclusion

---
**Require:**
   $\bar{\mathcal{S}} = \{(\boldsymbol{x}_i, \bar{y}_i)\}_i^N$: The given dataset;
**Ensure:** Classifier $f_\theta(\boldsymbol{x})$
 1: Randomly initialize a group parameter $\theta$ for $f_\theta(\boldsymbol{x})$;
 2: Randomly split $\bar{\mathcal{S}}$ into a training set $\bar{\mathcal{S}}_{\text{train}}$ and a valid-set $\bar{\mathcal{S}}_{\text{valid}}$;
 3: **for** $(e = 1; e \le Epochs; e + +)$ **do**
 4:   **for** $(\boldsymbol{x}_i, \bar{y}_i)$ $in$ $\bar{\mathcal{S}}_{\text{train}}$ **do**
 5:     $f_\theta(\boldsymbol{x}_i) = (\mu_1, \cdots, \mu_k)$;
 6:     $\mathbf{u} = \mathbf{1} - f_\theta(\boldsymbol{x}_i) = (1 - \mu_1, \cdots, 1 - \mu_k)$;
 7:     $loss = \ell(\mathbf{u}, \bar{y}_i)$;
 8:     $w = w - \beta \frac{\partial loss}{w}, w \in \theta$;
 9:   **end for**
10: **end for**
11: **return** $f_\theta(\boldsymbol{x})$

---

Together with some well known multi-class loss functions, such as CCE, MAE, MSE, the loss for learning with complementary labels with our definition are as follows:

$$\bar{\ell}(f_\theta(\boldsymbol{x}), \bar{y}) = \ell(\mathbf{v}, \bar{y}) = \begin{cases} \sum_{i=1}^k \mathbf{e}_{\bar{y}}^{(i)} \log \frac{1}{1-\mu_i} = \log \frac{1}{1-\mu_{\bar{y}}} & \text{CCE}, \\[2mm] \|\mathbf{e}_{\bar{y}} - \mathbf{v}\|_1 = k - 2 + 2\mu_{\bar{y}} & \text{MAE}, \\[2mm] \|\mathbf{e}_{\bar{y}} - \mathbf{v}\|_2^2 = k - 3 + \|\mathbf{u}\|_2^2 + 2\mu_{\bar{y}} & \text{MSE}, \end{cases} \quad (26)$$

where $\mathbf{e}_{\bar{y}}$ is a one-hot vector that the $\bar{y}$-th component equals to 1, others are 0. As its shown in Eq. (26), CCE and MAE loss satisfy the Theorem 1, MAE also satisfies the Theorem 2, while MSE does not satisfies the two. Zhang & Sabuncu (2018) propose a GCE loss function for learning with label noise, their formulation is as:

$$\ell_{\text{GCE}}(f_\theta(\boldsymbol{x}), y) = \frac{(1 - \mu_y^q)}{q}, \ q \in (0, 1). \quad (27)$$

It is easily to know that the loss function satisfies the constraint in Theorem 1, thus, it can be used to learning with complementary labels.

## 4.3 EXCLUSION ALGORITHM FOR LEARNING FROM COMPLEMENTARY LABELS

Based on the loss function we designed for complementary-label learning, we present an algorithm to learn a classifier from complementary labels with our loss function, named exclusion algorithm (the label specifies that the sample does not belong to it). The algorithm details show in Alg. 1. Furthermore, our algorithm is easily combined with the models designed for ordinary-label learning, with only a minus operation, which can be view as a framework to use the loss designed for ordinary-label learning to learn the optimal classifier from complementary labels.

## 5 EXPERIMENTS

## 5.1 EXPERIMENTAL SETTINGS

**Datasets.** We test our experiments on MNIST LeCun et al. (1998), FASHION-MNIST Xiao et al. (2017), CIFAR10 Krizhevsky (2009). Specifically, we generate two types of complementary labels: symmetric and asymmetric, for our experiments to verify our method's effectiveness and the theorem we proved in the previous section. For symmetric complementary-label, we fix a label distribution as Eq. (6) to generate the complementary-label training set. The validation set is split from the training set, which contains none ordinary-label. Thus, the lower the validation accuracy, the better the classifier learns from the training set. For asymmetric complementary-label, we randomly generate a matrix as Eq. (5) that the $p_{ij}$ is unknown as the complementary-label distribution and using it

to create complementary-label for experiments. The test accuracy of all experiments is tested on a clean dataset that contains only the ordinary labels.

**Approaches.** We test our loss with $\bar{\ell}_{CCE}, \bar{\ell}_{MAE}, \bar{\ell}_{MSE}, \bar{\ell}_{GCE}$ and compare with state-of-the-art methods in learning with complementary labels. The loss functions we used or compare in this paper are listed as follows. 1) CCE: The categorical cross-entropy loss, neither symmetric nor bounded, which widely use in machine learning and deep learning due to its fast convergence speed. 2) MAE: The mean absolute error, a symmetric loss and bounded, has been proved Ghosh et al. (2017) to be noise-tolerant. 3) MSE: The mean square error, not symmetric but bounded, widely used in regression learning. 4) GCE: It uses a hyper-parameter $q$ to tune the loss between MAE and CCE, but achieve noise-robust base on its bounded, we used the standard GCE where q=0.7 . 5) GA: Gradient ascent, a learning algorithm for complementary-label learning, is used to tackle the overfitting problem of the unbiased estimator they proposed in Ishida et al. (2019). 6) PC: Pairwise comparison (PC) with ramp loss designed for complementary-label learning Ishida et al. (2017). 7) Fwd: Forward correction Patrini et al. (2017), Yu et al. (2018a) designed for learning with complementary labels.

**Network architecture.** Following Ghosh et al. (2017), we use a network architecture that contains five layers to test the above methods for all the experiments: a convolution layer with 32 filters which filter size set as (3,3), a max-pooling layer with pooling-size of (3,3) and strides of (2,2), two fully connected layers with 1024 units, and a fully connected layer with soft-max activated function that the unit number set to the category number for prediction. Rectified Linear Unit (ReLU) is used as the activated function in the network's hidden layer.

**Implement details.** The implementation detail of our method shows in Alg. 1. We train our network with stochastic gradient descent through back-propagation. Each experiment trains 200 epochs, and the mini-batch size was set to 64. To exploit each loss function's best performance, we set three start learning rate for each loss function on each experiment and report the best accuracy amongst the three learning rate of each loss function. CCE is set to [1e-3, 5e-4, 1e-4], while GCE, MAE, MSE is set to [1.0, 0.5, 0.1]. The learning rate was halved per 50 epochs.

## 5.2 EXPERIMENTAL RESULTS

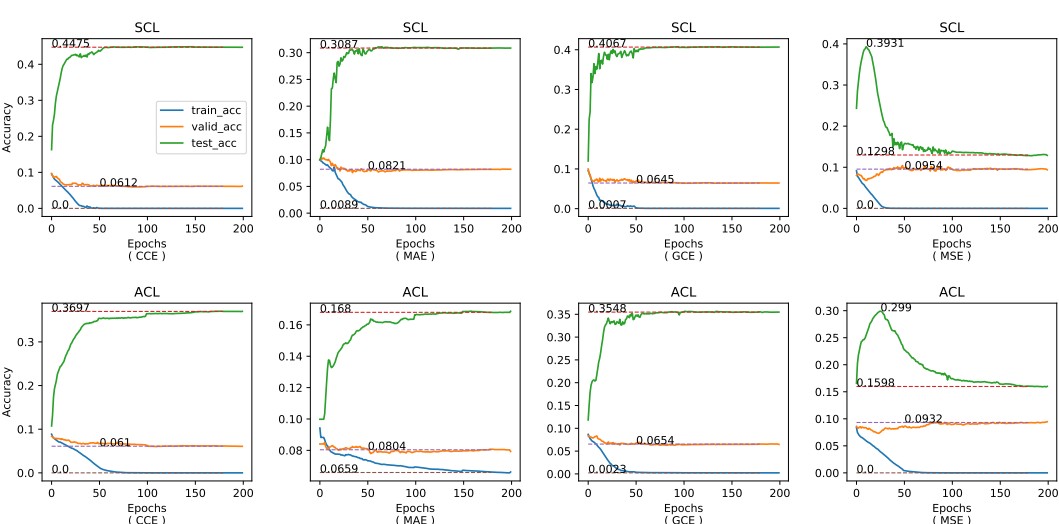

Figure 1: Accuracy for CCE, MAE, GCE, MSE loss functions over epochs, for CIFAR10 dataset with symmetric complementary labels (SCL) and asymmetric complementary labels (ACL). Legends are shown in the first sub figures on the first row.

**Robustness.** As shown in Fig. 1, together with CCE, MAE, and GCE loss, our algorithm achieves strong robust to both symmetric and asymmetric complementary labels, which verify that the robustness we prove in the Theorem 1 and Theorem 2. Even though the MAE satisfies the two theorems,

Table 1: The test accuracy and standard deviation (5 trials) on experiments with loss functions, under different complementary labels' distribution assumption, for datasets: MNIST, FASHION-MNIST, CIFAR10. We report the last ten percent epochs average test accuracy. For fair comparison, the last three columns' data are directly copying from Table.2 in Ishida et al. (2019), where GA Ishida et al. (2019): Gradient Ascent, PC Ishida et al. (2017): Pairwise Comparison, Fwd Yu et al. (2018b): Forward correction. The top 2 accuracies are **boldface**.

| Dataset | Distribution | Loss | | | | | | |
|---------|--------------|------|------|------|------|------|------|------|
| | | CCE | MAE | GCE | MSE | GA | PC | Fwd |
| MNIST | Symmetric | **95.66 ± 0.15** | 93.78 ± 3.66 | **97.46 ± 0.06** | 91.58 ± 0.60 | 88.1 ± 2.5 | 79.3 ± 3.3 | 88.7 ± 0.3 |
| | Asymmetric | **94.93 ± 0.12** | 68.11 ± 5.92 | **97.22 ± 0.12** | 85.98 ± 0.38 | – | – | – |
| FASHION | Symmetric | **86.43 ± 0.24** | 74.25 ± 0.26 | **86.43 ± 0.30** | 82.93 ± 0.18 | 78.7 ± 1.4 | 74.7 ± 1.6 | 77.5 ± 1.2 |
| | Asymmetric | **85.22 ± 0.19** | 54.01 ± 6.24 | **85.55 ± 0.12** | 78.93 ± 0.22 | – | – | – |
| CIFAR10 | Symmetric | **44.46 ± 0.31** | 27.78 ± 2.28 | **42.64 ± 0.82** | 36.10 ± 1.23 | 36.8 ± 0.6 | 33.4 ± 2.0 | 30.8 ± 1.6 |
| | Asymmetric | **37.93 ± 0.70** | 16.73 ± 0.22 | **36.01 ± 0.96** | 30.98 ± 0.74 | – | – | – |

it achieves a lower test accuracy than that of CCE and GCE due to it treats all labels the same (not sensitive to the labels). The subfigures in the last column of Fig. 1 shows that the MSE loss firstly achieves its highest test accuracy and then drop sharply over the epochs. Because MSE does not satisfy one of the two theorems we prove, it easily overfits the training set's complementary labels. Such a trend is the same as asymmetric complementary labels learning. The results verify that the algorithm we design for the complementary labels is significant and confirms the theoretical results we analyzed in the previous section.

**Performance Comparison.** The first four columns of Table. 1 show that the CCE and GCE loss achieve the best two test accuracies in our algorithm. In the MNIST dataset, the CCE achieves a little lower test accuracy than GCE, the same test accuracy in FASHION-MNIST, and a little higher test accuracy in CIFAR10 due to the dataset more challenge and CCE is more sensitive to labels. Even MAE is robust to complementary labels, and its performance is not well than others because it is a linear loss that is not sensitive to labels. As shown in Fig. 1, MSE is not robust to complementary labels, but with a small learning rate of 0.1, MSE only exhibited slight overfitting in Table 1. Furthermore, as shown in Table 1, together with CCE and GCE loss, our algorithm achieves a test accuracy higher than 95% in the MNIST dataset, which is comparable to that of learning with ordinary labels.

For a fair comparison, The last three columns directly form Ishida et al. (2019) even those results are the max test accuracy. In the first two datasets, all loss functions with our algorithm achieve a higher test accuracy than GA, but they used an MLP model as their base model, simpler than ours. In CIFAR10, they used ResNet-34 (21.62M parameters) He et al. (2016) and DenseNet Huang et al. (2017) as their based model, which is much bigger than ours (8.43M parameters), but we achieve a much higher test accuracy than theirs. The results validate the superiority of our algorithm to current state-of-the-art methods.

# 6 CONCLUSION

This paper designs an algorithm for learning from complementary labels using the loss functions designed for ordinary-label learning. We provide theoretical analysis to show that the loss functions we design for learning from the complementary labels are robust to the complementary labels, i.e., the optimal classifier learned from the complementary labels can theoretically converge to the optimal classifier learned from ordinary labels. In this paper, the two theorems we present are the sufficient condition of a loss function robust to complementary labels. Experimental results show that though complementary-label learning is a new topic in supervised-learning, it offers excellent competitiveness. More methods should be studied to improve the performance of complementary learning in our future works, such as Amid et al. (2019b) and Amid et al. (2019a).

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
