# OpenReview forum: "Robust Loss Functions for Complementary Labels Learning"
_ICLR.cc/2021/Conference — Reject_

### Official Review · AnonReviewer4 · 2020-10-27
**In need of some clarification, as well as a positioning with respect to learning with partial labels**

**Rating:** 3
**Confidence:** 4

**Review:**

This paper concerns the problem of learning from single-label supervision, when this label is known not to be the truth. This is called complementary label learning. Some loss functions are proposed that are claimed to have the same theoretical minimizer as the one for standard labelling.

The research agenda of the paper looks reasonable, even if it can be seen as a very specific instance of partial label learning (where one just considers the complement of the complementary label and tries to learn from it). A positioning with this latter approach therefore seems necessary. Also, after reading the paper, there are some unclarities left about the authors claim. Below are some more specific comments about that:

* Introduction: it is claimed that getting complementary label is easier than getting true labels, however complementary labels have to be certainly false, and while there are indeed theoretically more wrong labels than right ones, it is not entirely clear whether getting certainly false labels is easier than getting true ones in practice. Are there applications or empirical studies demonstrating that? Most mentioned papers do not appear to have actually applied the setting.

* Connection to partial label learning: the current framework can be seen as a peculiar case of partial label learning, as if I take a complementary label $\overline{y}$, then its complement $\mathcal{Y}\setminus\overline{y}$ is a partial label certainly containing the truth. It would then be necessary to connect the current work to this trend, for instance to Cour et al. "Learning from partial labels" (JMLR 2011) or the more recent works of, e.g., X Wu, ML Zhang "Towards Enabling Binary Decomposition for Partial Label Learning.".

* Definition 2: I do not really follow definition 2. First, are \theta^* and \theta arbitrary parameters values? Why call it \theta^* (suggesting some kind of optimality)? If   \theta^* is a minimizer of one of the two losses, then either the premise or the conclusion is a tautology (making the definition kind of meaningless).

* Proof of Theorem 1: I have some trouble with this definition. First, Equation (13) seems trivial if \theta^* is the finite sample optimal model (also, why not identifying the search space with the space of parameters?). I also do not really follow the next line, as it is unclear how realistic it is to modify the parameters for just one instance? It is also unclear what is to be proven here, as \inf \sum \geq \sum \inf, thus allowing for instance-specific parameters would always give something better than a global minimizer. In summary, I am not really convinced by this proof.

* In the experiment, I wold expect a comparison with other approaches (complementary but also partial label learning), but more importantly with the optimal models obtained on learning from the initial true labels, if only to demonstrate that the proposed theorems are valid. The asymptotic accuracies displayed are also very far from state-of-art standards (less than half of it) for CIFAR 10, which seems to contradict the fact that complementary labels have the same minimizer (hence comparable performances) as the one obtained with true labels?

* Finally, the paper contains an important numbers of typos or questionable grammatical structures. For instance in the first two pages only:
- "supper" --> super
- "A complementary-label is only specific that the pattern"
- "in some questions refer to private."
- "the best hyper-parameter by empirical risk since" (by empirical risk minimisation)
- "can be summary as"

---

> ### Author Response · Authors · 2020-11-16
> **Robust Loss Functions for Complementary Labels Learning**
>
> 1. Answer: Thank you very much for your question. To our best knowledge, complementary-label learning was first proposed by Ishida et al., 2017 ([A]). Lately, complementary-label learning has been applied to many applications:
> •	Online learning (Kaneko et al., 2019)
> •	Generative discriminative learning (Xu et al., 2019)
> •	Medical image segmentation (Rezaei et al., 2019)
> This suggests applying complementary labels to other domains may be useful, such as collecting survey data, which can be an attractive future direction.
> [A]: Ishida et al., 2017, "Learning from complementary labels."
> 2. Answer: Thank you very much for your suggestion. We have updated our introduction to connecting to partial label learning. “… partial label learning~\cite{cour2011learning, feng2018leveraging, wu2018towards} and others.”
> 3. Answer: Thank you very much for your question. Maybe it's a little unclear. $\theta$ is the arbitrary parameters in the searching space, $\theta^{\ast}$ is the parameters of a minimizer of risk in the training set. The difference between the two losses is that the training set is completely different. One contains complementary labels; another contains ordinary labels.
> 4. Answer: Thank you very much for your comments. Maybe it's a little unclear. In our paper, $\theta^{\ast}$ is the finite samples optimal model, $f$ represents the network architecture, and $\theta$ is the parameters. In Eq.(14), we prove by contradiction, to indicate that the optimal model $f_{\theta^{\ast}}$ supporting the Eq.(15). We have updated our paper, adding more details.
> 5. Answer: In the MNIST dataset, we achieved a higher than 95% test accuracy in the complementary-label dataset near the ordinary-label dataset. But in a more complex dataset, such as in cifar10, learning with the complementary-label dataset is far from learning with the ordinary-label dataset since a single ordinary label corresponds to (K-1) complementary labels. It needs more training data and research on complementary-label learning. Moreover, the methods designed for ordinary labels cannot be directly applied to the complementary labels.
> 6. Answer: Thank you very much for your comments. We have updated the paper carefully in a new version.
> •	"supper"--> super
> •	"A complementary-label is only specific that the pattern" --> A complementary Label is only indicating that the class label of a sample is incorrect.
> •	"in some questions refer to private." -->" For example, on some privacy issues, it is much easier to collect complementary labels than ordinary labels."
> •	"..can be summarized as.."

---

### Official Review · AnonReviewer1 · 2020-10-28
**An OK paper. Exposition needs to be improved. Interesting Theory. Results not fully convincing.**

**Rating:** 5
**Confidence:** 4

**Review:**

Summary:
=======
This paper deals with the problem of complementary label learning, that is, when we know the set of labels which a given observation does not belong to. In particular, the paper proposes a robust loss function and an algorithm for learning from complimentary labels. Results shown on MNIST and CIFAR datasets indicate the superior accuracy using the proposed loss function.


Comments:
==========
The paper addresses an important problem but it is written in a hurry which makes it hard to assess its contribution. There are many typos and other writing issues in the paper. The experiments are also weak. Though, the theoretical results are interesting and improve the previous known results for complementary label learning along certain dimensions.


1). Typically, in ML robust loss function means a loss function that is robust to outliers, e.g., the Huber Loss. However, the definition of robustness of loss function is different in this paper. However, in this paper it means if the loss function with ordinary and complementary labels has the same minimizer. I am unaware of this definition of robustness of a loss function as it seems very specific to the complementary label learning problem.


2). The results in Table 2 are not an apples-to-apples comparison. The numbers for GA, PC, Fwd are copied directly from other papers. In order to be fair, they should also similar base models as the authors. For instance, GA used used MLP which is less complex than the model used by the authors. So, it is unclear whether the improved performance is due to the difference in base architecture or due to the proposed robust loss function.


Typos:

Page 1: "However, label such a large-scale dataset is time-consuming..."
Page 1: "In the view of label noise, complementary labels can also be view as..."
Page 3: "...only complementary labels that specific the samples does not..."
Many others!

---

> ### Author Response · Authors · 2020-11-16
> **Robust Loss Functions for Complementary Labels Learning**
>
> 1. Answer: Thank you for your question. There may be something we do not express very clear. In ML, a loss function is called noise-tolerant if the minimizer of risk (under that loss function) with noisy labels would be the same as that with noise-free labels [A]. In our definition, when the complementary labels are viewed as the noise labels, there is no difference between our definition and the ML robust loss function.
> [A]. Ghosh, A., Kumar, H., & Sastry, P. S. (2017). Robust loss functions under label noise for deep neural networks.
> 2. Answer: Thank you for your comment.  As shown in Table.1, in the first two datasets (MNIST, FASHION), we use a complex model than the MLP model, and our method achieved higher performance than others in the prior work. Surely, the improved performance is unclear. But in CIFAR10, we use a simpler model than ResNet-34 they used and achieved higher performance. In this scenario, it is no doubt that the improved performance is due to our method.
> 3. Answer: Thank you very much for your comments. We have updated the paper carefully in a new version.
> Page 1: "However, labeling such a large-scale dataset is time-consuming..."
> Page 1: "In the view of label noise, complementary labels can also be viewed as..."
> Page 3: "...only labels indicating that the class label of a sample is incorrect. "

---

> > ### Comment · AnonReviewer1 · 2020-11-20
> > **Thank you.**
> >
> > I have read the authors' rebuttal and they have answered my questions.  I have increased my score. Though, I am still on the margin regarding this paper, especially after reading some of the comments from other reviewers regarding the connection of this work with "Partial Labels" literature.

---

### Official Review · AnonReviewer3 · 2020-10-28
**Simple yet insightful conditions on loss and simple training algorithm for successful complementary label learning**

**Rating:** 7
**Confidence:** 3

**Review:**

The paper presents (two) simple yet insightful sufficient conditions for a usual loss to work well as a complementary label loss. Based on this, a simple training procedure, which minimally differs from usual training, is presented. Empirically it is shown that the proposal outperforms state-of-the-art.

Comments:
1. I like the simple idea of CL-loss (8). Also the analysis 4.2, though simple, is elegant and insightful. For example, it helps to identify that MSE is not an ideal CL-loss.
2. The improvement in performance over GA/PC/Fwd is impressive in table1.
3. Overall, the write-up is well-organised; however at places I felt the presentation can be simplified a lot. sometimes it is because of grammar, sometime because of confusing notation, and sometimes because of too much reading between lines. For e.g., since (16) is an important step, more explanation seems necessary (notation further complicates the ease), e.g., in (26),(27) one-step reasons for verifying validness/non-validness as cl-loss may be helpful.
4. Though conditions (10)/(20) are insightful for determining robustness, for a reader who is encountering them for the first time, it may help to intuitively explain the conditions.
5. In fig1 why does training accuracy decrease with epochs?

---

> ### Author Response · Authors · 2020-11-16
> **Robust Loss Functions for Complementary Labels Learning**
>
> 1. Answer: Thank you for your comment.
> 2. Answer: Thank you for your comment.
> 3. Answer: Thank you very much for your helpful comments. We added more explanation in Eq.(16) to help to understand in a new version: “According to Eq.~(\ref{sufficient_condition}), $\bar{\ell}$ is a monotone increase loss function only on $\mathbf{u}^{(\bar{y})}$, then we have …”
> 4. Answer: Thank you very much for your helpful comments.  In Eq.(10), it means that $\bar{\ell}$ is a monotone increasing loss function \textbf{only on} $\mathbf{u}^{(\bar{y})}$. In Eq.(20), it means that $\bar{\ell}$ is a symmetric loss ($\sum \ell(\mathbf{u}, i)=C$), and $\bar{\ell}$ is a monotone increasing loss function on any $\bar{y}$.
> We have updated in a new version.
> 5. Answer: In complementary-label learning, there are no ordinary labels in the validation set and the training set. Our goal is to train a classifier to predict the ground-true label for any sample drawn from the same distribution as the training set. So, the training accuracy decrease with epochs.

---

### Official Review · AnonReviewer2 · 2020-10-29
**An interesting work for complementary label learning.**

**Rating:** 7
**Confidence:** 2

**Review:**

This paper studied a new problem, that is, learning from complementary labels.  The goal is to predict a correct label for a given sample when only given complementary labels. On the basis of the ordinary-label learning, the authors defined "robust loss functions" for complementary-label learning:  a a loss function is called robust  loss function if minimizer of risk with complementary labels would be the same as with ordinary  labels. Then, they provided  two sufficient conditions for the robust loss function and a exclusion algorithm is provided for prediction. Experimental results show that the proposed method outperforms other methods in several datasets.

Overall, the problem is interesting and important, and the proposed algorithm seems reasonable and effective. However, I think the paper could be improved from the following two aspects.

1.I suggest to add ablation test to give a deep analysis about the effectiveness of the proposed algorithm.

2.In Section 5.1, it is better to give more explanations about why the lower the validation accuracy, the better the classifier learns from the training set.

3. What is the exact form of complementary label? Suppose there are k classes, or k labels. If we know that a sample does not belong to given k-1 classes, then we can deduce that it belongs to the rest one class.   So, it is important to specify the exact form of complementary label to show the necessity of complementary-label learning.

---

> ### Author Response · Authors · 2020-11-16
> **Robust Loss Functions for Complementary Labels Learning**
>
> 1. Answer: Thank you very much for your suggestion. We are studying the two methods in [A] and [B], and considering a more effective model to improve the performance of complementary learning.
> We have updated the conclusion: More methods should be studied to improve the performance of complementary learning in our future works, such as [A] and [B].
> [A]: Amid, Ehsan, Manfred K. Warmuth, and Sriram Srinivasan. "Two-temperature logistic regression based on the Tsallis divergence." The 22nd International Conference on Artificial Intelligence and Statistics. PMLR, 2019.
> [B]: Amid, Ehsan, et al. "Robust bi-tempered logistic loss based on bregman divergences." Advances in Neural Information Processing Systems. 2019.
> 2. Answer: Thank you very much for your question. In complementary-label learning, there are no ordinary labels in the validation set and the training set. Our goal is to train a classifier to predict the ground-true label for any sample drawn from the same distribution as the training set. So, the lower the validation accuracy, the better the classifier learns from the training set.
> 3. Answer: Thank you for your helpful suggestion. The form of a complementary label is that a sample only assigned a label that specifies the class it does not belong to. For example, $(x, 1)\in \bar{\mathcal{S}}$, it means that sample $x$ does not belong to $class-1$.

---

### Public Comment · ~Ehsan_Amid1 · 2020-11-10
**Please consider referencing/comparing to these more recent works**

I would like to point out that our work (Amid et al. 2019a) extends the Generalized CE loss (Zhang and Sabuncu 2018) by introducing two temperatures t1 and t2 which recovers GCE when t1 = q and t2 = 1. Our more recent work, called the bi-tempered loss (Amid et al. 2019b) extends these methods by introducing a proper (unbiased) generalization of the CE loss and is shown to be extremely effective in reducing the effect of noisy examples. Please consider referencing/comparing to these papers.

(Amid et al. 2019a) Amid et al. "Two-temperature logistic regression based on the Tsallis divergence." In The 22nd International Conference on Artificial Intelligence and Statistics (AISTATS), 2019.

(Amid et al. 2019b) Amid et al. "Robust bi-tempered logistic loss based on Bregman divergences." In Advances in Neural Information Processing Systems (NeurIPS), 2019.

---

> ### Author Response · Authors · 2020-11-16
> **Robust Loss Functions for Complementary Labels Learning**
>
> Thank you for your helpful suggestion. We have referenced the two papers and updated our paper.

---

### Decision · Program_Chairs · 2021-01-07
**Final Decision**

**Decision:**

Reject

**Comment:**

the paper undoubtedly tackles an interesting problem in the mainstream of learning with partial / unknown / weak / noisy / complementary labels. The authors have had a set of constructive suggestions and questions from the reviewers (and external comments), some positive, some negative. I find it a bit unsettling that to several major questions, the main feedback from the authors was a citation in the paper with no further action; (a) ablation tests of R2 end up in citing papers from a public comment, (b) R4 raised a key point in comment 2, with the links to partial labels learning. The authors’ answer is not satisfying as one would have hoped at least of a partial justification of the author’s approach in this context. The authors would have had time to develop at least elements of a formal comparison. Just citing the work is not enough;